# Conditional cash transfers and mortality in people hospitalised with psychiatric disorders: A cohort study of the Brazilian Bolsa Família Programme

Camila Bonfim[1]*, Flávia Alves[1,2], Érika Fialho[1], John A. Naslund[2]
Maurício L. Barreto[1], Vikram Patel[2], Daiane Borges Machado[1,2]

1 Centre of Data and Knowledge Integration for Health (CIDACS), Fiocruz-Bahia, Salvador, Bahia, Brazil,
2 Department of Global Health and Social Medicine, Harvard Medical School, Boston, Massachusetts, United States of America

* camila.bonfim@fiocruz.br

## Abstract

### Background

Psychiatric patients experience lower life expectancy compared to the general population. Conditional cash transfer programmes (CCTPs) have shown promise in reducing mortality rates, but their impact on psychiatric patients has been unclear. This study tests the association between being a Brazilian Bolsa Família Programme (BFP) recipient and the risk of mortality among people previously hospitalised with any psychiatric disorders.

### Methods and findings

This cohort study utilised Brazilian administrative datasets, linking social and health system data from the 100 Million Brazilian Cohort, a population-representative study. We followed individuals who applied for BFP following a single hospitalisation with a psychiatric disorder between 2008 and 2015. The outcome was mortality and specific causes, defined according to International Classification of Diseases 10th Revision (ICD-10). Cox proportional hazards models estimated the hazard ratio (HR) for overall mortality and competing risks models estimated the HR for specific causes of death, both associated with being a BFP recipient, adjusted for confounders, and weighted with a propensity score. We included 69,901 psychiatric patients aged between 10 and 120, with the majority being male (60.5%), and 26,556 (37.99%) received BFP following hospitalisation. BFP was associated with reduced overall mortality (HR 0.93, 95% CI 0.87,0.98, $p$ 0.018) and mortality due to natural causes (HR 0.89, 95% CI 0.83, 0.96, $p$ < 0.001). Reduction in suicide (HR 0.90, 95% CI 0.68, 1.21, $p$ = 0.514) was observed, although it was not statistically significant. The BFP's effects on overall mortality were more pronounced in females and younger individuals. In addition, 4% of deaths could have been prevented if BFP had been present (population attributable risk (PAF) = 4%, 95% CI 0.06, 7.10).

**Data Availability Statement:** The code used in the analysis is available from Github [https://github.

com/profacamilabonfim/Codes-from-the-paper.git] and archived in Zenodo [https://zenodo.org/records/13750552]. The data analyzed in this study is hosted by the Centre of Data and Knowledge Integration for Health (CIDACS). Full access to the data is restricted due to its sensitive nature and the exclusive licensing agreement for its use in this study. The privacy regulations set by the Brazilian Ethics Committee prohibit the public availability of this data. However, upon reasonable request, and provided that all ethical and legal requirements are met, the institutional data curation team can make the data available. Further information can be obtained by emailing cidacs.curadoria@fiocruz.br. Study Protocol available on: https://pubmed.ncbi.nlm.nih.gov/36201469/.

**Funding:** This study was supported by funding from the National Institute of Mental Health (Grant number: 5R01MH128911, awarded to DBM). The funder had no role in the design of the review; collection, analysis, or interpretation of data; writing of the manuscript; or decision to submit the manuscript for publication.

**Competing interests:** The authors have declared that no competing interests exist.

**Abbreviations:** ATT, average treatment effect on the treated; BFP, Bolsa Família Programme; CCTP, conditional cash transfer programme; CI, confidence interval; HR, hazard ratio; ICD-10, International Classification of Diseases 10th Revision; IPTW, inverse probability of treatment weighting; KM, Kernel matching; LMIC, low- and middle-income country; PAF, population attributable risk; PS, propensity score; SIPTW, stabilised inverse propensity scores; SMD, standardised mean difference.

## Conclusions

BFP appears to reduce mortality rates among psychiatric patients. While not designed to address elevated mortality risk in this population, this study highlights the potential for poverty alleviation programmes to mitigate mortality rates in one of the highest-risk population subgroups.

---

## Author summary

### Why was this study done?

- People living with psychiatric disorders have a higher risk of mortality compared to the general population.
- Poverty contributes to these individuals experiencing more risky behaviours and receiving less healthcare.
- Conditional cash transfer programmes (CCTPs) have shown an association with reduced mortality in the general population; however, there is a lack of studies investigating this among psychiatric patients.

### What did the researchers do and find?

- We performed a population-based cohort study to investigate the association between the Brazilian Bolsa Família Programme (BFP) and the risk of overall mortality and specific causes, such as natural and unnatural causes, as well as suicide among psychiatric patients.
- We observed that BFP recipients had lower mortality rates when compared to nonrecipients, especially for overall mortality and natural causes of death.
- In addition, we found that a considerable number of deaths could have been prevented if this benefit had been present.

### What do these findings mean?

- These are the first results to suggest that a broad programme of social assistance, not directed at psychiatric patients in particular, has major benefits in reducing mortality among psychiatric patients after discharge.
- It has broader significance too, opening the question of whether such assistance could reduce the well-established high mortality rate among all people with psychiatric disorders.
- The main limitation of this study is that the results are not generalizable to all hospitalised individuals, as we have accessed severe cases that were hospitalised in public services only. However, these services in Brazil cover 75% of the population.

## Introduction

Epidemiological studies show that individuals with psychiatric disorders have a shortened life expectancy [1–3]. Compared to the general population, the mortality rate is nearly double among those living with psychiatric disorders [2,4] and 80% higher among mental health service users [5]. This mortality gap is predicted to worsen, as reflected in recent Global Burden Disease study data, demonstrated by the rise in psychiatric disorders from 13th position to the seventh leading cause of disability-adjusted life-years over the past decade [3].

Individuals with psychiatric disorders commonly face multimorbidity [6], reflected by co-occurring chronic medical conditions such as cardiovascular diseases, respiratory illnesses, diabetes, hepatitis, and obesity [7]. Moreover, this patient population often receives poor quality healthcare, including limited access to effective mental health services, a concern that is particularly severe in low- and middle-income countries (LMICs) [8]. These individuals face numerous barriers to accessing health services, such as transportation difficulties, lengthy waiting lists, language and cultural barriers, and experiences of stigma and discrimination [9]. Inadequate care, prevalent in LMICs, is closely linked to poor physical health outcomes and increased mortality [10,11].

Research has also demonstrated variations in mortality rates among individuals with psychiatric disorders. While the relative risks for unnatural causes of death, such as violence, suicide, road injuries, and falls, were elevated among individuals with psychiatric disorders, when compared to natural causes of death, such as cardiovascular and respiratory diseases and cancer, over two-thirds of deaths in this population group may be attributed to natural causes [12]. We highlight that suicide rates are higher for individuals hospitalised with psychiatric disorders when compared to those in primary care settings, reflecting added vulnerabilities among those seeking care in a hospital environment [13,14]. Although suicide is a multifactorial problem, research shows that people with psychiatric disorders experience a higher risk of suicide, compared to the general population. Psychiatric disorders may increase the risk of suicide 10-fold [15]; it is estimated that approximately 62.2% of the global burden of suicide may be associated with psychiatric disorders [16] even though these global estimates rarely include LMICs. The differences in mortality rates may be influenced by multiple factors, including genetic, behavioural, aspects of lifestyles, access to healthcare, psychiatric treatments, and social determinants of health, such as poverty and lack of social support [12].

There is a vicious cycle between poverty and psychiatric disorders, with longitudinal studies showing that conditional cash transfer programmes (CCTPs) in LMICs, in addition to alleviating poverty, may reduce the burden of mental health problems [17–19], suicidal behaviour [20,21], and premature mortality [22,23]. Cash transfer programmes are social policies commonly implemented in LMICs with the main goal of alleviating extreme poverty and poverty-related outcomes such as food insecurity. These programmes can exhibit considerable heterogeneity among countries in terms of transfer value, frequency and duration of the benefit, presence of conditionalities such as health and schooling attendance, targeting and eligibility criteria, as well as implementation systems (government or private) [24,25].

Therefore, cash transfer programmes have been associated with multiple additional benefits, such as improving financial security and family stability and reducing financial strain, which are factors commonly associated with natural causes of death, such as cardiovascular diseases [22] and unnatural causes of death, such as suicide [20] and violence [26]. In addition, CCTPs include conditions, such as improving primary care services and regulating school attendance, which may improve beneficiary health behaviour [27], thereby contributing to reduced mortality rates [23,28]. Despite these promising tendencies, the potential benefits of CCTPs have not yet been evaluated among the vulnerable population of individuals

hospitalised with psychiatric disorders. Therefore, the contribution of CCTPs to this population would be particularly relevant, considering that there is an increased risk of death after discharge [29].

Very few studies have explored the potential impact of economic interventions, such as CCTPs, on natural and unnatural causes of mortality among individuals hospitalised with psychiatric disorders. These insights are valuable for supplementing policies to promote the health and longevity of individuals with psychiatric disorders, while advancing broader efforts to address the disproportionately elevated mortality risk affecting this population group, which has been identified as a major priority for global mental health [3,9,30]. We hypothesise that the Bolsa Família Programme (BFP) could reduce the mortality risk among psychiatric patients. The objective of this study was to test the association of participating in a CCTP through Brazil's national BFP, and the risk of mortality due to overall, natural, and unnatural causes, as well as suicide in those previously hospitalised with any psychiatric disorders.

## Methods

### Study design, data sources, and dataset linkage

We conducted this evaluation according to Machado and colleagues' protocol [31]. This cohort study uses data from the 100 Million Brazilian Cohort, linked to the Hospitalisation (SIH) and Mortality Information Systems (SIM) (2001–2018) [31] (S6 Text). The 100 Million Brazilian Cohort is a dynamic cohort comprised of individuals registered on CadÚnico, which is the primary system used when applying for social assistance in Brazil [32]. CadÚnico is a database for selecting and including families in the various government-supported social programmes, which is linked with BFP database. It enables families in situations of socioeconomic vulnerability to access these benefits. Covering roughly 55% of the total Brazilian population, it comprises individuals facing poverty and extreme poverty [31,32].

Brazil presents one of the world's largest and most complex public health systems, ranging from primary to tertiary levels, the latter providing hospital care. Access to this system is comprehensive, universal, and free for the entire population. Although a private supplementary system exists, over 70% of the Brazilian population relies on the public health system, particularly the poorest segments of society [33].

SIH encompasses 75% of Brazil's total admissions in general or specialised hospitals under this national health system [34]. It focuses on serious morbidity requiring specialised attention and monitoring, often involving advanced resources [35]. SIM records all Brazilian deaths through mandatory, high-quality death certificates [36]. Both systems use standardised forms completed by health professionals, including the cause of a hospital admission and death, according to the International Classification of Diseases 10th Revision (ICD-10) [31].

The databases were connected using nondeterministic linkage and a tool developed by a team of experts at the Centre of Data and Knowledge Integration for Health (CIDACS)/Fiocruz, to link administrative data from Brazil [37] (S1 Text). Further information on data governance and the linkage process has been published elsewhere [37,38]. Ethical approval for the study was obtained from the Federal University of Bahia (UFBA—registration number: 1023107). This study is reported as per STROBE guideline (S1 STROBE Checklist).

### Participants

Our study population comprised individuals aged 10 and older who had been registered on the cohort baseline at different times and had at least 1 hospitalisation with a psychiatric disorder (defined by code "F", according to ICD-10 [39], and registered on SIH) between January 1, 2008 and December 31, 2015 when all the data were available.

We identified all individuals aged 10 and older who had been first hospitalised with psychiatric disorders in the study period. We analysed all the hospital discharge records of patients with primary or secondary diagnoses classified as a psychiatric disorder over the same period. We included secondary diagnoses, considering psychiatric disorders that were also common among people admitted with other primary diagnoses [40]; however, this represented only 5.07% of the total admissions. Cutoff at the age of 10 is justified since suicide, one of the outcomes investigated in this study, is an extremely rare occurrence under this age [20]. Moreover, considering that individuals with a history of multiple hospitalisations due to psychiatric disorders are at a higher risk of mortality [41], we only included individuals with a single hospitalisation in our analysis in order to enhance comparability among participants. In our data, approximately 3% of individuals were readmitted for psychiatric hospitalisation. Subsequently, we selected all individuals registered on CadÚnico following their first hospitalisation in this period to avoid selection bias. Beneficiaries might be less frequently hospitalised than nonbeneficiaries, considering the association between reduction of poverty and improved health conditions [22,23,28], causing an imbalance with the comparison group. A description of the individuals excluded from the analysis can be found in the Supporting information (S2 Text). Finally, we excluded individuals who had anomalous information that could reflect linkage errors (Fig 1) (S2 Text).

For the BFP beneficiary subset, these individuals were followed from the time they registered to receive the BFP benefit, and their follow-up ended either due to the individual's death by any cause, or on December 31, 2015 (reflecting the end of the follow-up period). For the nonbeneficiary subset (i.e., individuals who were not registered for the BFP benefit), the follow-up started when these individuals were registered on CadÚnico. The follow-up ended for nonbeneficiaries either due to their death by any cause, or the end of the follow-up period on December 31, 2015.

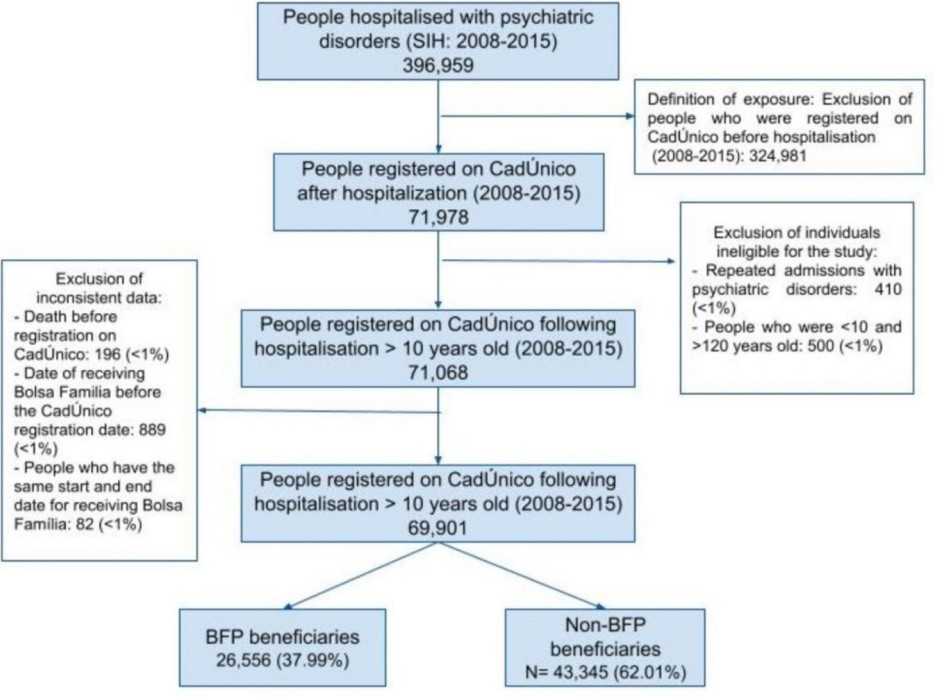

**Fig 1. Flowchart of the study population.** BFP, Bolsa Família Programme.

## Exposure and covariates

BFP is a conditional cash transfer that targets the poorest families [42], with conditions related to healthcare for children and pregnant women, and children's education. Eligibility is based on CadÚnico system registrations and having a monthly household income of less than BRL 70.00 (USD 17.00), or BRL 140.00 (USD 34.00) for households with a child, adolescent, or pregnant woman. Benefits range from BRL 41.00 (USD 10.00) to a maximum of BRL 300.00 (USD 75.00) per person, utilising 2015 values, adjusted for inflation [32]. Conditions for receiving the BFP benefit include school attendance, vaccinations, and monitoring young people's growth. Pregnant or breastfeeding women must also follow a health and nutrition protocol [32]. We assumed that individuals receiving BFP were considered exposed throughout the study, covering benefit receipt and adherence to conditions for receiving BFP, impacting their quality of life and health outcomes [32]. While beneficiaries can stop receiving BFP due to noncompliance or increased income, 95% remained in the programme until the last year of the follow-up period. The study compared the group receiving BFP benefits with those who did not, defining beneficiaries as individuals receiving benefits during the follow-up. In addition, only those registered for BFP posthospitalisation with psychiatric disorders were included, focusing on the effect of BFP registration following this type of hospitalisation and reducing potential biases related to better socioeconomic conditions that could improve mental health and reduce hospitalisation rates [29].

Covariates were defined based on a literature review and selected from those registered on databases [14–20]. This included sex, age, race, education level, location of residence (rural or urban), household characteristics (presence of a water supply, waste, sanitation, and construction materials), crowding (number of people in the house by the number of rooms), isolation (people who live alone, or with someone else), Brazilian regions of residence, year of hospitalisation, year of CadÚnico registration, and length of hospitalisation. The latter was calculated in terms of days and categorised based on tertiles, considering the first admission record. Most of these covariates were registered at the cohort baseline, except for hospitalisation-related variables, which were registered at the time of a hospital admission.

## Outcomes

This study included overall, natural, and unnatural causes of death, as well as suicide, recorded on SIM between 2008 and 2015. Natural causes were defined by all causes of mortality, according to ICD-10 [39], except for external causes, using violence and suicide codes. Unnatural causes included external causes, such as accidents, falls, suicide, or violence (V010 –Y98) [39], while the overall causes of mortality included all general medical conditions. Furthermore, suicide was analysed separately (codes X60 to X84) [35], given the strong association between suicide and psychiatric disorders [15].

## Statistical analyses

Descriptive analyses were carried out to characterise the demographic, socioeconomic, and hospitalisation covariates, comparing BFP beneficiaries to nonbeneficiaries (Table 1). Standardised mean differences (SMDs) were computed before and after inverse probability of treatment weighting (IPTW) estimation, and absolute values higher than 0.1 indicated imbalance in the distribution of covariates between treated and untreated groups (Table 1) [20]. Mortality rates were estimated using person-years as the denominator for each individual's observation period (S1 Table).

In line with the study protocol [31], and other studies using the 100 Million Brazilian Cohort [20,23,27,28], the propensity score (PS)–based method was applied to promote

**Table 1. Description of BFP nonbeneficiaries and beneficiaries before and after IPTW, 2008 to 2015.**

| Characteristics | Before IPTW (N = 69,901) | | | After IPTW (N = 57,905) | | |
|---|---|---|---|---|---|---|
| | BFP N = 26,556 (37.99) | Non-BFP N = 43,345 (62.01) | Diff. [1] (BFP – non-BFP) | BFP N = 20,399 (35.22) | Non-BFP N = 37,506 (64.78) | Diff.[1] (BFP – non-BFP) |
| | % | % | | % | % | |
| Sex | | | | | | |
| Male | 56.92 | 62.70 | 0.21 | 0.52 | 0.53 | 0.03 |
| Female | 43.08 | 37.30 | | 0.48 | 0.47 | |
| Age group (years old) | | | | | | |
| 10-24 | 12.11 | 7.36 | −0.39 | 0.13 | 0.12 | −0.31 |
| 25-59 | 84.23 | 80.49 | | 0.83 | 0.84 | |
| >60 | 3.66 | 12.14 | | 0.04 | 0.04 | |
| Education Level (years of education) | | | | | | |
| Never studied | 10.43 | 15.74 | 0.10 | 0.10 | 0.11 | −0.01 |
| Preschool | 0.76 | 1.41 | | 0.01 | 0.07 | |
| Primary school or less (<=5 years) | 34.87 | 35.86 | | 0.35 | 0.34 | |
| Junior high school (6-10 years) | 27.37 | 20.57 | | 0.28 | 0.27 | |
| High school (10-12 years) | 24.86 | 23.91 | | 0.25 | 0.25 | |
| College/university (>=13 years) | 1.42 | 2.23 | | 0.01 | 0.01 | |
| Missing data | 0.29 | 0.28 | | NA | NA | |
| Race | | | | | | |
| White | 43.06 | 47.91 | 0.08 | 0.47 | 0.47 | 0.01 |
| Black | 8.88 | 6.84 | | 0.09 | 0.09 | |
| Asian | 0.41 | 0.51 | | 0.01 | 0.01 | |
| Mixed race/Brown | 41.50 | 39.16 | | 0.44 | 0.44 | |
| Indigenous | 0.23 | 0.10 | | 0.01 | 0.01 | |
| Missing data | 5.93 | 5.49 | | NA | NA | |
| Location of residence | | | | | | |
| Rural | 9.06 | 8.17 | 0.05 | 0.91 | 0.92 | 0.01 |
| Urban | 81.18 | 90.95 | | 0.09 | 0.08 | |
| Missing data | 9.76 | 0.88 | | NA | NA | |
| Brazilian regions | | | | | | |
| Southeast | 49.09 | 43.83 | −0.10 | 0.49 | 0.49 | −0.01 |
| Northeast | 14.57 | 13.70 | | 0.15 | 0.14 | |
| Central-West | 6.89 | 10.27 | | 0.07 | 0.07 | |
| South | 27.28 | 29.74 | | 0.27 | 0.28 | |
| North | 2.17 | 2.45 | | 0.02 | 0.02 | |
| Household characteristics | | | | | | |
| Water supply | | | | | | |
| Public network (running water) | 71.57 | 83.77 | 0.13 | 0.87 | 0.87 | 0.03 |
| Well, natural sources, or other | 12.75 | 9.60 | | 0.13 | 0.13 | |
| Missing data | 15.68 | 6.63 | | NA | NA | |
| Waste | | | | | | |
| Public collection system | 77.41 | 88.10 | 0.09 | 0.93 | 0.93 | 0.02 |
| Burned, buried, outdoor disposal, or other | 6.91 | 5.28 | | 0.07 | 0.07 | |
| Missing data | 15.68 | 6.63 | | NA | NA | |
| Sanitation | | | | | | |

(*Continued*)

**Table 1.** (Continued)

| Characteristics | Before IPTW (N = 69,901) | | | After IPTW (N = 57,905) | | |
|---|---|---|---|---|---|---|
| | BFP N = 26,556 (37.99) | Non-BFP N = 43,345 (62.01) | Diff.[1] (BFP – non-BFP) | BFP N = 20,399 (35.22) | Non-BFP N = 37,506 (64.78) | Diff.[1] (BFP – non-BFP) |
| | % | % | | % | % | |
| Public network | 54.63 | 61.74 | 0.04 | 0.67 | 0.68 | 0.02 |
| Septic tank | 9.94 | 11.91 | | 0.12 | 0.12 | |
| Homemade septic tank | 13.46 | 16.25 | | 0.16 | 0.16 | |
| Ditch or other | 3.60 | 1.94 | | 0.05 | 0.04 | |
| Missing data | 18.36 | 8.16 | | NA | NA | |
| Construction materials | | | | | | |
| Bricks/cement | 70.85 | 80.78 | 0.04 | 0.85 | 0.85 | 0.01 |
| Wood, other plant materials, or other | 13.47 | 12.59 | | 0.15 | 0.15 | |
| Missing data | 15.68 | 6.63 | | NA | NA | |
| Isolation | | | | | | |
| Lives with someone else | 66.58 | 73.16 | −0.16 | 0.78 | 0.76 | −0.01 |
| Lives alone | 33.42 | 26.84 | | 0.22 | 0.24 | |
| Year of registration on CadÚnico | | | | | | |
| 2008 | 2.75 | 0.62 | −0.55 | 0.03 | 0.03 | −0.05 |
| 2009 | 7.50 | 1.72 | | 0.09 | 0.09 | |
| 2010 | 11.59 | 3.74 | | 0.15 | 0.14 | |
| 2011 | 11.50 | 10.63 | | 0.13 | 0.13 | |
| 2012 | 19.58 | 25.66 | | 0.20 | 0.21 | |
| 2013 | 17.60 | 17.53 | | 0.13 | 0.13 | |
| 2014 | 17.20 | 20.79 | | 0.16 | 0.16 | |
| 2015 | 12.28 | 19.31 | | 0.11 | 0.11 | |

BFP, Bolsa Família Programme; IPTW, inverse probability of treatment weighting.

[1]The difference in proportions of each category between BFP beneficiaries and nonbeneficiaries (BFP beneficiary proportion minus nonbeneficiary proportion).

comparability between treated and untreated groups. We estimated the PS using a multivariable logistic regression, adjusted for socioeconomic covariates associated with BFP [43] (S2A Table). The following covariates were considered when estimating the PS: sex, age, education level, race, location of residence (rural or urban), household characteristics (presence of a water supply, waste, sanitation, and construction materials), crowding (number of people in the house by the number of rooms), isolation (people who live alone or with someone else), Brazilian regions of residence, and year of CadÚnico registration. The last item was included considering variations in the CadÚnico register over time. Additionally, we assessed the common support graph, and we compared the range of propensity scores among BFP and non-BFP groups (S3 Fig and S2B Table).

Then, we estimated the IPTW [44,45]. IPTW uses the PS to balance differences among covariates in the treated and untreated groups using weights [45]. Individuals who received the BFP were given weights equal to the inverse of their propensity scores (1/PS), whereas those who did not receive the treatment were given weights equal to the inverse of 1 minus their propensity scores [PS/(1 −PS)] [44,45]. Following this, we identified problems with extreme weights in the estimation. To correct this problem, truncated weights were used with specified thresholds based on weight distribution for the 99th percentile [27]. To estimate the

association between BFP and mortality rates, we used an average treatment effect on the treated (ATT) estimator, according to the recommendation of Ali and colleagues [45] and other studies using 100 million Brazilian cohort [20,22,23,27,28], and fitted as a survival analysis model using Cox proportional hazard regression [46]. Hazard ratio (HR) was estimated for the overall mortality with 95% confidence intervals (CIs), and we introduced IPTW weights as a weight function in Stata, in addition to the year and length of the hospitalisation, to control for mortality-related risk factors. We also estimated a competing risks model using a Fine Gray model, which directly models the subdistribution hazard for each cause of death, accounting for the presence of competing risks [47]. In each competing risk model, we considered each cause of death as the failure, the other causes as competing risks, and the individuals who were alive were censored (Table 2). Participants with missing covariates data were excluded from the final model.

Furthermore, to better identify the potential effects of a public policy as BFP on reducing mortality, we calculated an equivalent population attributable risk (PAF) to estimate the proportion of death that BFP theoretically could prevent [48]. We used the formula PAF = [P(HR − 1)] / [P(HR − 1) + 1] [48], where "P" was the incidence of the BFP in the population and "HR" is the hazard ratio estimated for the association between BFP and overall mortality. We estimated unadjusted and adjusted PAF using the punafcc package in Stata, which uses Cox regression and a 95% CI according to previous publication [49]. For the adjusted model, we included the same covariates as in the final model.

**Table 2. Association of BFP participation with overall, natural, unnatural, and suicide mortalities, 2008–2015.**

| Confounder adjustment | Overall population | Cox model | Competing risks model | | |
|---|---|---|---|---|---|
| | | Overall mortality | Natural causes | Unnatural causes | Suicide |
| | | HR (95% CI) | HR (95% CI) | HR (95% CI) | HR (95% CI) |
| Cox adjusted with IPTW[1] (final model) | | | | | |
| Non-BFP | 57,905 | 1.00 | 1.00 | 1.00 | 1.00 |
| BFP | | 0.93 (0.87, 0.98) | 0.89 (0.83, 0.96) | 1.14 (0.97, 1.33) | 0.90 (0.68, 1.21) |
| *p*-value | | 0.018 | 0.001 | 0.112 | 0.514 |
| Sensitivity analysis | | | | | |
| Cox adjusted with SIPTW[2] | | | | | |
| Non-BFP | 57,905 | 1.00 | 1.00 | 1.00 | 1.00 |
| BFP | | 0.91 (0.86, 0.97) | 0.87 (0.82, 0.93) | 1.21 (1.04, 1.40) | 0.94 (0.71, 1.23) |
| *p*-value | | 0.002 | <0.001 | 0.012 | 0.642 |
| Cox adjusted with kernel matching[3] | | | | | |
| Non-BFP | 57,475 | 1.00 | 1.00 | 1.00 | 1.00 |
| BFP | | 0.77 (0.72, 0.81) | 0.74 (0.69, 0.78) | 1.02 (0.89, 1.18) | 0.91 (0.69, 1.20) |
| *p*-value | | <0.001 | <0.001 | 0.795 | 0.502 |

BFP, Bolsa Família Programme; CI, confidence interval; HR, hazard ratio; IPTW, inverse probability of treatment weighting; SIPTW, stabilised inverse propensity scores.

[1]HR estimated with IPTW given sex, age, race, education level, household characteristics (water supply, waste, sanitation, and construction materials), living alone, crowding, Brazilian region, location of residence, length and year of hospitalisation, and year of CadÚnico registration.

[2]HR estimated with SIPTW given sex, age, race, education level, household characteristics (water supply, waste, sanitation, and construction materials), living alone, crowding, Brazilian region, location of residence, length and year of hospitalisation, and year of CadÚnico registration.

[3]HR estimated with K matching given sex, age, race, education level, household characteristics (water supply, waste, sanitation, and construction materials), living alone, crowding, Brazilian region, location of residence, length and year of hospitalisation, and year of CadÚnico registration.

**Table 3. Association of BFP participation with overall, natural, unnatural, and suicide mortalities by subgroups, 2008–2015.**

| Subgroups | Cox model | Competing risks model | | |
|---|---|---|---|---|
| | Overall mortality | Natural causes | Unnatural causes | Suicide |
| | HR[1] (95% CI) | HR[1] (95% CI) | HR[1] (95% CI) | HR[1] (95% CI) |
| Sex | | | | |
| Male | 1.05 (0.98, 1.13) | 1.02 (0.94, 1.11) | 1.24 (1.04, 1.48) | 0.86 (0.61, 1.21) |
| N | 33,786 | 33,786 | 33,786 | 33,786 |
| *p*-value | 0.169 | 0.650 | 0.016 | 0.391 |
| Female | 0.75 (0.67, 0.85) | 0.73 (0.64, 0.83) | 0.91 (0.63, 1.35) | 1.02 (0.56, 1.82) |
| N | 24,119 | 24,119 | 24,119 | 24,119 |
| *p*-value | <0.001 | <0.001 | 0.666 | 0.928 |
| Age groups (years old) | | | | |
| 10- 24 | 0.79 (0.57, 1.10) | 0.56 (0.33, 0.95) | 1.21 (0.79, 1.85) | 0.84 (0.33, 2.08) |
| N | 5,589 | 5,589 | 5,589 | 5,589 |
| *p*-value | 0.174 | 0.033 | 0.387 | 0.708 |
| 25-59 | 0.95 (0.89, 1.01) | 0.91 (0.84, 0.98) | 1.16 (0.98, 1.39) | 1.01 (0.73, 1.37) |
| N | 47,261 | 47,261 | 47,261 | 47,261 |
| *p*-value | 0.131 | 0.015 | 0.080 | 0.992 |
| 60 or older | 0.97 (0.83, 1.11) | 0.98 (0.84, 1.14) | 0.90 (0.43, 1.89) | 0.18 (0.02, 1.29) |
| N | 5,135 | 5,135 | 5,135 | 5,135 |
| *p*-value | 0.554 | 0.839 | 0.794 | 0.087 |

BFP, Bolsa Família Programme; CI, confidence interval; HR, hazard ratio; IPTW, inverse probability of treatment weighting.

[1]HR estimated with IPTW given sex, age, race, education level, household characteristics (water supply, waste, sanitation, and construction materials), living alone, crowding, Brazilian region, location of residence, length and year of hospitalisation, and year of CadÚnico registration.

We used multiple approaches for the sensitivity analysis. First, we employed the stabilised inverse propensity scores (SIPTW) approach, a modification of IPTW that stabilises weights to enhance numerical stability during estimation processes. We estimated SIPTW weights for nonbeneficiaries using the formula $(1 - Pt) / (1 - Psmul)$, and for beneficiaries using the formula $Pt / Psmul$, where "Pt" represents the marginal probability of treatment in the population, and "Psmul" denotes the propensity score obtained from multivariable logistic regression adjusted for covariates. We applied the same truncation process to extreme weights and repeated the final model. Second, we employed the Kernel matching (KM) approach, which establishes a weighting scheme for all untreated units, assigning greater weights to units closer to those treated units to which they are matched. This method matches pairs based on weights estimated from propensity scores. Beneficiaries and nonbeneficiaries were matched by year of registration in the cohort, and the PS matched with kernel weights for the same covariates used in the final model generating the ATT (S3A–S3D Table). Third, to assess the relevance of IPTW in obtaining unbiased BFP estimates, we conducted crude and adjusted Cox regressions without IPTW weighting (S4 Table). Fourth, the final model was repeated, using missing values as a category in the analysis (S5 Table). Fifth, subgroup analyses were performed by sex and age using IPTW weights, calculated for each subcategory (Table 3). Finally, we repeated these analyses using Poisson models (S6 Table). Stata version 15.0 was used for data analysis.

## Results

We identified 369,959 individuals hospitalised with psychiatric disorders on the SIH database between 2008 and 2015. When these individuals were linked to the CadÚnico database

after their first hospitalisation in this period, we identified 71,978 individuals who entered the 100 Million Brazilian Cohort. We then excluded 910 participants (<1%) who did not meet the inclusion criteria. Finally, we excluded 1,167 participants (<1%), due to inconsistent data. The study sample only included 69,901 individuals who had applied for BFP following a single hospitalisation with any psychiatric disorders and met the eligibility criteria (Fig 1).

Twenty-six thousand, five hundred and fifty-six (26,556) (37.99%) individuals who had been hospitalised with psychiatric disorders received BFP. The average time of receiving BFP after discharge was 2.86 years (SD = 1.85). Before IPWT weighting, there were differences in sociodemographic characteristics between beneficiaries and nonbeneficiaries (Table 1). BFP beneficiaries, compared to nonbeneficiaries, were more likely to be aged between 25 and 59 (84.23% versus 80.49%), non-white (56.94% versus 52.09%), live in the Southeast region of Brazil (49.09% versus 43.83%), and reside in more crowded households (0.79 versus 0.57), respectively. In contrast, when comparing nonbeneficiaries to beneficiaries, they were more likely to be male (62.70% versus 56.92%), live in urban areas (90.95% versus 81.18%), and not live alone (73.16% versus 66.58%), respectively (Table 1). There was a progressive increase in CadÚnico registration over the period studied, while psychiatric hospitalisations decreased for both groups, and approximately half (52.81%) of all hospitalisations were for more than 2 weeks (S6 Text). After IPTW weighting, the beneficiary and nonbeneficiary groups had similar sociodemographic characteristics. The difference in proportions of each category between BFP beneficiaries and nonbeneficiaries was lower than 10% (Table 1).

Over the period, 8,118 individuals died for overall causes and most of them were nonbeneficiaries (63.5%) (S1 Table). Mortality rates per 100,000 person-years for natural causes (non-BFP 4,077.98 95% CI 3,958.81, 4,200.74 versus BFP 2,783.79 95% CI: 2,672.95, 2,899.23) were higher when compared to unnatural causes (non-BFP 738.65 95% CI 688.93, 791.96 versus BFP 757.26 95% CI 700.51, 818.61). For suicide, the rates were 176.49 95% CI 153.04, 203.53 for non-BFP and 144.75 95% CI 121.13, 172.98 for BFP recipients (S1 Table).

The subgroup analysis observed lower mortality rates among females for all causes, particularly in beneficiaries. Among individuals aged between 10 and 24, mortality rates were lower among hospitalised nonbeneficiaries, compared to beneficiaries for natural causes (416.28 versus 755.17 per 100,000 person-years) and overall mortality (1,185.49 versus 1,546.89 per 100,000 person-years), respectively. For those aged 60 or older, mortality rates due to unnatural causes were lower for hospitalised nonbeneficiaries compared to beneficiaries (647.74 versus 665.08 per 100,000 person-years). Suicide rates were lower across different age groups, especially among hospitalised nonbeneficiaries aged between 10 and 24 (158.34 per 100,000 person-years) and beneficiaries aged 60, or older (39.12 per 100,000 person-years; S1 Table).

BFP was associated with a reduction in overall mortality (HR 0.93; 95% CI 0.87, 0.98; $p = 0.018$) and mortality due to natural causes (HR 0.89; 95% CI 0.83, 0.96; $p = $ <0.001; Table 2). The associations between BFP beneficiares and mortality rates were similarly observed in sensitivity analyses and after including missing covariate values as missing categories. The effects of BFP appeared strongest among females and younger individuals (Table 3). When clustering by household level was accounted for, the estimate of the effect of BFP on overall mortality was null (IRR 0.98; 95% CI 0.92, 1.04; $p = 0.478$) (S4 Text and S7A and S7B Table).

The unadjusted PAF analysis showed that 18% (95% CI 16, 20) of deaths could potentially be prevented if BFP had been present, while the adjusted analysis showed a reduction of 4% (95% CI 0.06, 7.10).

## Discussion

To our knowledge, this is the first study to estimate the association of a CCTP with mortality in individuals hospitalised with psychiatric disorders registered on the 100 Million Brazilian Cohort. BFP was associated with a 7% reduction in the overall mortality rate among beneficiaries, primarily driven by lower mortality due to natural causes. For mortality due to unnatural causes and suicide, in particular, results were consistent with an effect, but they were not statistically significant. Furthermore, the effects of BFP were strongest among females and younger individuals. Furthermore, 4% of these deaths could be prevented if BFP were present.

Previous studies have demonstrated how CCTPs contribute to breaking the bidirectional cycle of poverty and psychiatric disorders in the general population [50]. However, less is known about how these programmes could break this cycle among those already affected by severe psychiatric disorders. Therefore, this study contributes to understanding the role of a CCTP in increasing the chance of survival in a population subgroup that disproportionately faces financial hardship and complex mental and physical health care needs. These findings illustrate the potential of BFP in advancing tertiary prevention within this highly vulnerable patient population.

This study highlights the potential impact of BFP in reducing mortality rates among those hospitalised with psychiatric disorders, attesting to the importance of implementing social protection programmes to cover vulnerable population subgroups. Although BFP was not specifically designed to address health and social concerns affecting individuals with psychiatric disorders, the programme appears to have important downstream benefits on their health and on reducing mortality rates. The BFP focus on various aspects of health, coupled with poverty alleviation, may have assisted in facilitating access to primary care services and routine check-ups for individuals with psychiatric disorders, thereby resulting in improvements in their health behaviour and, ultimately, reducing natural causes of death [22]. The strong association between receiving BFP and a reduction in natural causes of death suggests a synergistic effect between BFP and the Family Health Strategy [51,52], whereby individuals supported by BFP experience direct health benefits through preventive measures and management of comorbidities, such as hypertension, diabetes, and other chronic diseases. Over time, through increased access to basic and preventive health services, BFP may have had a positive influence on alleviating the elevated mortality rates affecting individuals with psychiatric disorders.

In Brazil, the public healthcare system for individuals with psychiatric disorders is provided through the Psychosocial Care Network, which includes services such as primary care, community-based mental health centers (called psychosocial care centers, or CAPS, in Brazil), emergency and urgent care network, residential services, and general hospitals [53]. The Brazilian health reform has increasingly prioritized mental health care in the CAPS, progressively reducing psychiatric hospital beds, which were the primary option before the reform [54]. However, there is a shortage of these services, with better access found in larger cities, despite some limited expansion into small municipalities [55]. In cases where these services are not available, general hospitals may be the only option for individuals experiencing a psychiatric emergency, such as those living with severe mental disorders. Although the provision of mental health care in general hospitals can bring benefits such as stigma reduction, increased service access, improved physical health care, and the possibility of multidisciplinary team care, the number of psychiatric beds available in these hospitals remains limited [54]. Therefore, these findings may also reflect weaknesses in access to urgent mental health services through the psychosocial care network.

A stronger effect of BFP on reducing mortality due to natural causes and overall mortality rates was observed for women and the younger population. This is consistent with previous

research, which found that BFP reduces overall mortality, especially among women and younger populations [22,56]. Although most of the sample was composed of men, BFP had a greater impact on reducing mortality rates among women. This can potentially be explained, given that BFP emphasises women's important role, and the benefit is provided to women who are, for the most part, heads of households [57]. Previous studies have demonstrated that CCTPs may encourage female empowerment and facilitate decision-making power among women in relation to managing household budgets [57].

Although these findings showed that BFP was associated with an overall reduction in overall and natural cause mortality rates in individuals hospitalised with psychiatric disorders, the association between receiving BFP benefits and a reduction in mortality due to unnatural causes or suicide did not emerge as being statistically significant. There are some possible reasons for these unexpected findings. First, the sample size and lower frequency of deaths due to unnatural causes and suicide could have affected the power of the study, thereby reducing the probability of detecting statistically significant differences between BFP beneficiaries and non-beneficiaries [58]. Second, unnatural causes of death may be influenced by multifactorial aspects, such as behaviour and the social environment [59], which would require a longer follow-up period to fully observe any benefits that could be attributed to BFP within this specific subgroup.

This study has a number of limitations. While death certificates are mandatory in Brazil, and SIM is recognised for its high-quality standards [36], underreporting is always a possibility. Psychiatric disorders require a comprehensive clinical response, with services delivered by mental health professionals. Although the diagnosis was based on ICD-10, one of the leading classifications for mental health diagnoses, we lacked the means to validate the accuracy of this diagnosis [60]. In addition, the datasets did not provide information on the severity of the psychiatric disorders. However, given that psychiatric hospitalisation is typically reserved for severe cases when outpatient resources have been exhausted [61], it was hypothesised that most cases in this study involved severe mental disorders requiring hospital support. Consequently, these findings cannot be generalised for patients with mild mental disorders who receive care in primary and secondary outpatient services.

The data used in this study are not generalisable to all hospitalised individuals since the system only covers public service information and may be influenced by service availability. Nonetheless, as recorded on CadÚnico, this study population has limited access to private health insurance and cannot afford private healthcare services. Despite this limitation, the system is considered suitable for conducting epidemiological research, since it captures approximately 75% of hospitalisations in Brazil [34]. Also, there are possible issues with missing information and diagnosis misclassification [34]. However, improvements have been made in the quality of SIH records in recent years [34], thereby mitigating this concern.

This study used an administrative database that was not designed for research purposes. Thus, there were a number of issues with missing values, mainly in variables that are not mandatory in the system. However, main variables, such as cause of mortality, diagnosis, individual information (e.g., sex and age), and access to benefits, were mostly complete. Furthermore, covariates were only measured at the cohort baseline and were not updated at a later date. Moreover, the linkage process may generate a bias due to challenges such as computational complexity and the absence of a unique number that may identify health and social systems. However, sensitivity and specificity exceeding 94% was achieved in the linkage validation process, and these errors are likely to be nondifferential (S1 Text). An additional limitation is the potential bias related to unmeasured confounding, especially socioeconomic and behavioural factors that were not available in the routinely collected datasets. Finally, data cannot be generalised for the entire Brazilian population and only reflects the poorest segments since the

database only includes those seeking social benefits. Compared to the Brazilian population, this cohort overrepresents young people and women due to the BFP target population [32]. Furthermore, the profile of individuals hospitalised for psychiatric reasons in our study also differed from other studies [54]. This might have happened because in our study, the selection criteria focused exclusively on individuals who registered at CadÚnico after hospitalisation. This criterion was intentionally chosen to reduce potential biases related to better socioeconomic conditions that could improve mental health and reduce hospitalisation rates. Consequently, this may have resulted in a cohort that differs from those in other studies where different selection criteria were employed.

This study is highlighted as the first, to our knowledge, to examine the impact of BFP on mortality rates among individuals hospitalised with psychiatric disorders. Using a robust analysis such as competing risk models, this study stands out by showing how the BFP reduces the overall risk of death, and its impact on specific cause of death, considering the competing risks of death from other causes. These findings reveal a noteworthy effect, indicating that receiving financial assistance intended for poverty alleviation can potentially reduce the mortality risk in this vulnerable population subgroup. These results underscore the importance of considering intersectoral strategies for tertiary prevention posthospitalisation of mental health patients. Collaborative initiatives with BFP not only contribute to financial stability but possibly also address institutional barriers, thereby playing a pivotal role in shaping mortality outcomes among individuals hospitalised with psychiatric disorders. The large size of the cohort allowed an evaluation of mortality due to specific causes among BFP nonbeneficiaries and beneficiaries in this specific population, hospitalised with psychiatric disorders. The large size of the cohort also made it possible to explore the effects of BFP on less common health outcomes and the effect variation among subgroups.

This study also contributes to current knowledge on the role of a large economic intervention in alleviating the elevated mortality risk among people hospitalised with severe psychiatric disorders in a LMIC. BFP has primarily reduced overall and natural causes of mortality, especially for women and young people who are commonly the target of social policies due to several poverty-related vulnerabilities. Importantly, BFP has provided economic support and demonstrated the potential to act as a tertiary prevention intervention. Thus, the results in this study have important practice and policy implications to advance efforts for early mortality prevention in mental health care settings and provide support for this vulnerable population group facing the challenges of serious psychiatric disorders.

## Supporting information

**S1 Strobe Checklist. STROBE Statement—Checklist of items that should be included in reports of observational studies.**
(DOCX)

**S1 Text. Accuracy analysis of the linkage between CadÚnico and the mortality information system in a randomised sample of 10,000 record pairs.**
(DOCX)

**S2 Text. Detailed information from eligible study population.**
(DOCX)

**S3 Text. Propensity score: Definition, estimation, summary, and support graphs.**
(DOCX)

**S4 Text. Intraclass correlation coefficient estimation.**
(DOCX)

**S5 Text. Description of individuals excluded from the analysis.**
(DOCX)

**S6 Text. Summary of the dataset's description.**
(DOCX)

**S1 Fig. ROC curve of the 100 million Brazilian Cohort and SIH (2001–2018) linkage.**
Source: Developed by the CIDACS Data Production Center.
(TIF)

**S2 Fig. ROC curve of the 100 million Brazilian Cohort and SIM (2000–2015) linkage.**
Source: Developed by the CIDACS Data Production Center.
(TIF)

**S3 Fig. Distribution of the propensity score in the sample, 2008–2015.**
(TIF)

**S1 Table. Mortality rates overall and by subgroups through receipt of the BFP, 2008–2015.**
(DOCX)

**S2 Table. (A) Logistic regression to estimate propensity scores for receiving Bolsa Familia according to covariables, *N* = 57,905 (B) Propensity score description in accordance with the confounding covariates observed, Brazil, 2008 to 2015, *N* = 57,905.**
(DOCX)

**S3 Table.** (**A**) ATT of overall mortality for BFP receipt between 2008 and 2015 using KM. (**B**) ATT of natural causes of death for BFP receipt between 2008 and 2015 using KM. (**C**) ATT of unnatural causes of death for BFP receipt between 2008 and 2015 using KM. (**D**) ATT of suicide for BFP receipt between 2008 and 2015 using KM.
(DOCX)

**S4 Table. Crude and adjusted association of BFP participation with overall, natural, unnatural, and suicide mortalities, 2008–2015.**
(DOCX)

**S5 Table. Association of BFP participation with overall, natural, unnatural, and suicide mortalities accounting for missing data, 2008–2015.**
(DOCX)

**S6 Table. Incidence rate ratio of BFP participation with overall, natural, unnatural, and suicide mortalities, 2008–2015.**
(DOCX)

**S7 Table.** (**A**) Intraclass correlation estimation for the household level. (**B**) Association of BFP participation with overall mortality considering household level, 2008–2015.
(DOCX)

**S8 Table. Description of individuals excluded from the analysis following definition of BFP exposition, 2008–2015.**
(DOCX)

**S9 Table. Description of year and length of hospitalisation overall and by BFP participation, 2008–2015.**
(DOCX)

## Acknowledgments

We thank the data production team and all CIDACS/FIOCRUZ collaborators for their work on building the 100 Million Brazilian Cohort. We also thank Kosuke Imai for his invaluable assistance and expertise in providing statistical support for this research.

The content is solely the authors' responsibility and does not necessarily represent the official views of the National Institutes of Health.

## Author Contributions

**Conceptualization:** Daiane Borges Machado.

**Data curation:** Camila Bonfim.

**Formal analysis:** Camila Bonfim, Érika Fialho.

**Funding acquisition:** Daiane Borges Machado.

**Investigation:** Daiane Borges Machado.

**Methodology:** Daiane Borges Machado.

**Project administration:** Daiane Borges Machado.

**Supervision:** Maurício L. Barreto, Vikram Patel, Daiane Borges Machado.

**Validation:** Flávia Alves, Érika Fialho.

**Visualization:** Flávia Alves, Érika Fialho, John A. Naslund.

**Writing – original draft:** Camila Bonfim.

**Writing – review & editing:** Flávia Alves, John A. Naslund, Maurício L. Barreto, Vikram Patel, Daiane Borges Machado.

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
