## [Editor Report · Decision Letter 0]

3 Apr 2024

Dear Dr Bonfim, 

Thank you for submitting your manuscript entitled "Do conditional cash transfers reduce mortality in people hospitalised for psychiatric disorders? A quasi-experimental analysis of the Brazilian Bolsa Família Programme" for consideration by PLOS Medicine.

Your manuscript has now been evaluated by the PLOS Medicine editorial staff and I am writing to let you know that we would like to send your submission out for external peer review.

Please re-submit your manuscript within two working days, i.e. by 5th April. If you need extra time, please do let me know.

Kind regards,

Syba Sunny, MBBS, MRes, FRCPath

Associate Editor

PLOS Medicine

ssunny@plos.org

---

## [Decision Letter · Decision Letter 1]

11 Jun 2024

Dear Dr. Bonfim,

Many thanks for submitting your manuscript "Do conditional cash transfers reduce mortality in people hospitalised for psychiatric disorders? A quasi-experimental analysis of the Brazilian Bolsa Família Programme" (PMEDICINE-D-24-01060R1) for consideration at PLOS Medicine. 

Your paper has now been reviewed by 3 subject reviewers and a statistician; it was also discussed with an academic editor with relevant expertise and the wider editorial team. The comments are included below and can also be accessed here:

[LINK]

As you will see, the reviewers found your paper interesting. However, there were requests for further information and clarification of various points. In addition, the statistical reviewer (Reviewer 1) raised several important concerns about the study’s design and analysis. Nevertheless, we remain interested in your work. After discussing the paper with the editorial team, I’m pleased to invite you to revise the paper in response to the reviewers’ comments and our editorial requests (below). We plan to send the revised paper to some or all of the original reviewers*, and of course we cannot provide any guarantees at this stage regarding publication.

When you upload your revision, please include a point-by-point response that addresses all of the reviewer and editorial points, indicating the changes made in the manuscript and either an excerpt of the revised text or the location (e.g. page and line number) where each change can be found. Please submit a clean version of the paper as the main article file and a version with changes marked should as a marked-up manuscript. Please also check the guidelines for revised papers at http://journals.plos.org/plosmedicine/s/revising-your-manuscript for any that apply to your paper.

We ask that you submit your revision by July 2nd, 2024. However, if this deadline is not feasible, please contact me by email (ssunny@plos.org), and we can discuss a suitable alternative.

Please don’t hesitate to contact me directly with any questions (ssunny@plos.org). 

Kind regards,

Pippa - Senior Editor, PLOS Medicine - on behalf of:

Syba Sunny MBBS, MRes, FRCPath

Associate Editor 

PLOS Medicine

ssunny@plos.org

*Please note: If your article is accepted, you may have the opportunity to make the peer review history publicly available. The record will include editor decision letters (with reviews) and your responses to reviewer comments. If eligible, we will contact you to opt in or out.

EDITORIAL COMMENTS:

We thank you for submitting your valuable work to PLOS Medicine. We note your unique data set, but, equally, agree with all the points raised by the reviewers. As such, we require that you address all the reviewer and editorial comments in full, before we can consider your paper further.

1) Data Availability:

Please revise your data availability section. At present, there is missing information – please replace the parts with XXX with the relevant information. 

If the data are not freely available, please describe briefly the ethical, legal, or contractual restriction that prevents you from sharing it. Please also include an appropriate contact (web or email address) for inquiries (this cannot be a study author). Further information is available from

http://journals.plos.org/plosmedicine/s/data-availability

2) Reporting guidance:

Please report your study according to the relevant guidance which can be found here https://www.equator-network.org/reporting-guidelines/

3) Statistical reporting:

(Note that not all of the following instructions will apply to your work, but please review all points and action as appropriate.)

Please quantify the main results with 95% CIs and p values.

When reporting p values please report as <0.001 and where higher as p=0.002, for example. When reporting 95% CIs please separate upper and lower bounds with commas instead of hyphens as the latter can be confused with reporting of negative values.

Please include the actual amounts and/or absolute risk(s) of relevant outcomes (including NNT or NNH where appropriate), not just relative risks or correlation coefficients. (example for absolute risks: PMID: 28399126).

Please include any important dependent variables that are adjusted for in the analyses.

4) Prespecified analysis plan/study protocol:

Did your study have a prospective protocol or analysis plan? Please state this (either way) early in the Methods section.

For all observational studies, in the manuscript text, please indicate: (1) the specific hypotheses you intended to test, (2) the analytical methods by which you planned to test them, (3) the analyses you actually performed, and (4) when reported analyses differ from those that were planned, transparent explanations for differences that affect the reliability of the study's results. If a reported analysis was performed based on an interesting but unanticipated pattern in the data, please be clear that the analysis was data-driven.

5) Abstract layout:

Please structure your abstract using the PLOS Medicine headings (Background, Methods and Findings, Conclusions), i.e. combine the 2 middle sections together to make a section called ‘Methods and Findings’.

6) Author summary:

At this stage, we ask that you include a short, non-technical Author Summary of your research to make findings accessible to a wide audience that includes both scientists and non-scientists. The authors summary should consist of 2-3 succinct bullet points under each of the following headings:

• Why Was This Study Done? Authors should reflect on what was known about the topic before the research was published and why the research was needed.

• What Did the Researchers Do and Find? Authors should briefly describe the study design that was used and the study’s major findings. Do include the headline numbers from the study, such as the sample size and key findings. 

• What Do These Findings Mean? Authors should reflect on the new knowledge generated by the research and the implications for practice, research, policy, or public health. Authors should also consider how the interpretation of the study’s findings may be affected by the study limitations. In the final bullet point of ‘What Do These Findings Mean?’, please describe the main limitations of the study in non-technical language.

The Author Summary should immediately follow the Abstract in your revised manuscript. This text is subject to editorial change and should be distinct from the scientific abstract. Please see our author guidelines for more information: https://journals.plos.org/plosmedicine/s/revising-your-manuscript#loc-author-summary

7) Introduction layout:

Please address past research and explain the need for and potential importance of your study. Indicate whether your study is novel and how you determined that. If there has been a systematic review of the evidence related to your study (or you have conducted one), please refer to and reference that review and indicate whether it supports the need for your study.

8) Discussion layout:

Please present and organize the Discussion as follows: a short, clear summary of the article's findings; what the study adds to existing research and where and why the results may differ from previous research; strengths and limitations of the study; implications and next steps for research, clinical practice, and/or public policy; one-paragraph conclusion.

9) Miscellaneous:

Please revise your ‘Short Title’ – the word ‘condicional’ should be replaced with ‘conditional’.

Comments from the reviewers:

Reviewer #1: 

Using data from the 100 Million Brazilian Cohort, this paper investigates the effect of conditional cash transfers on mortality among people previously hospitalised for any psychiatric disorder. A propensity score-based approach to analysis is taken. I have several questions about the analysis and other statistical aspects of the study. 

1. This study is referred to as "quasi-experimental". However, I cannot determine what the experimental element of this study is. It would seem preferable to refer to this as an observational study.

2. Given that the patients included in the dataset for analysis may not have had a primary psychiatric disorder diagnosis, is it correct that all included patients were hospitalised due to a psychiatric disorder? It seems that some included patients were people hospitalised with a psychiatric disorder (rather than "for a psychiatric disorder").

3. In the analysis presented in Tables 3 and 4, overall mortality is considered, but then mortality is divided up by cause. However, if an individual dies due an unnatural cause, they cannot die due to some other cause (for example). Thus, the results of the analysis presented in Table 3 for all apart from overall mortality do not have a useful interpretation. An appropriate analysis would account for competing risks of death by other causes, using a competing risks model. 

4. It is unclear what the propensity score-based analysis that was applied here was - I could not access the Appendix, where additional details may have been supplied. What is clear is that a propensity score model for receipt of BFP was fit. However, the authors mention that IPTW was applied, but the details presented here do not align with a standard IPTW analysis. In such an analysis, all individuals would receive a weight of one over the probability that they received their actual "treatment" (here a cash transfer or not). Instead, here those participants who did receive the cash transfer had a weight of 1, and those participants who did not receive the cash transfer had another weight. I am not sure what is meant by "E(ps) is the probability if receiving the programme in the population": is this the average propensity score? The following approach is recommended:

a. Assess the common support condition by comparing the range of propensity scores in each exposure group, and exclude participants outside the range of common support.

b. Generate IPTWs following the formulas in reference 39.

c. Calculate standardised differences in the weighted space (i.e. after applying the IPTWs). It is impossible to determine how well the propensity score model has performed without the calculation of these standardised differences. 

d. If standardised difference have reduced sufficiently, then the weighted outcome regression model can be it. This will include the exposure as the single covariate. Including the IPTWs as a covariate is not a recommended approach. 

5. I do not understand how the ATT was calculated, how the stabilised versions of the IPTWs were calculated or how the kernel matching approach was applied. More details are required.

6. How many individuals were excluded due to missing data?

7. How were the standardised mean differences calculated for the categorical characteristics? Generally an SMD would be calculated for each level of the characteristic. 

8. Given my comments on the analysis above, it is difficult to determine whether the conclusions drawn in this study are supported by the data. 

Reviewer #2: 

This is an interesting article using a quasi-experimental study design to examine whether enrollment in a conditional cash transfer program was associated with reduced mortality among Brazilians previously hospitalized for a psychiatric illness. I am generally in agreement with the analysis and its interpretation. However, I did have the following questions:

1) Why was the sample restricted to those with a psychiatric hospitalization only once during the study period? Researchers provide a rationale but I'm unsure how does this differ from other studies that looked at the overall Brazilian Bolsa Família Programme enrollment and mortality risk from different causes.

2) Did you stratify according to whether the participant primary vs. secondary diagnosis was for psychiatric disease to see if results differ according to disease severity? 

3) Your propensity score did not control for any health-related covariates? Is health a strong predictor for enrollment in this CCT program?

4) Participants in your study only applied for the CCT program after their psychiatric hospitalization - would the initial hospitalization influence enrollment in CCT in any way?

5) Minor - I am confused when the researchers write about hospitalized beneficiaries in the text because I thought all the participants were hospitalized.

6) Minor - What is the rationale for the age groupings?

Reviewer #3: 

This manuscript reports findings from an interesting analysis on a large Brazilian database in which the authors investigated mortality after a psychiatric hospitalisation and investigated associations with conditional cash transfer payments. My comments are as follows: 

1. Line 121 - for a journal of this general nature, I think a sentence could usefully be added on what a CCTP entails, beyond simply saying that it's a social policy; in particular, some description is needed as to whether it's a tightly defined intervention internationally or is describing something much broader and heterogeneous. Currently the paper assumes that this will be familiar territory to all readers. 

2. Lines 146-157 - likewise, there are quite a lot of assumptions about readers knowledge of Brazilian healthcare. For interpretation, it would probably be important to be clear what healthcare options are available in Brazil and what proportion of people will register for social assistance (and thus, presumably, appear on the source dataset). 

3. Figure 1 - the substantial drop from n=152,862 to n=71,068 is a little concerning. Am I correct in assuming that 'registration date after follow-up has ended' represented an indication of data errors? If so, this seems quite a high level and it would be helpful if there was some assurance of data quality for the remainder. 

4. Is BFP registration at individual or household level? If at household level, was any procedure deployed to account for clustering?

5. Table 2 - is there any particular reason for the inconsistency in the ordering of columns between the two comparison groups?

6. Is there any information that can be provided on the extent of propensity score overlap between the two comparison groups? Is it appropriate to be including people with 0% or 100% propensity scores (assuming there were some of them)? Or do the sensitivity analyses cover this scenario?

7. As a general comment, for a paper whose analyses are focused on public health and policy, it would be worth considering the inclusion of metrics with stronger communication in that field. From the results available, wouldn't it be possible to include a PAF equivalent (i.e., the proportion of deaths that could theoretically have been prevented with the intervention, assuming causality) and/or a NNT equivalent (number receiving the intervention required to result in one fewer deaths over a given follow-up period)?

Reviewer #4: Manuscript Number: PMEDICINE-D-24-01060R1

This manuscript is based in the 100 Million Brazilian Cohort, a dynamic cohort representing people who registered for CadUnico, a system used to apply for social assistance in Brazil. The authors examined whether a conditional cash transfer benefit, the Bolsa Familia Program (BFP) reduced mortality after a single hospitalization for a primary or secondary psychiatric diagnosis during 2008-2015. Importantly, the conditions for receiving the BFP cash transfer include participation education, health and other social programs. 

The study sample comprised those who registered for CadUnico and applied for BFP after their psychiatric hospitalization and before 2015. Within the study sample, they compared mortality rates up to 2015 for those who received vs did not receive BFP. Those who received BFP were followed from the time they received it, and those who did not receive BFP were followed from the time they registered for CadUnico. BFP was associated with a substantially reduced mortality rate overall. This was primarily driven by a reduced mortality for "natural" causes, although there was also a trend for "unnatural" causes such as suicide. 

These are the first results to suggest that a broad program of social assistance, not directed at psychiatric patients in particular, has major benefits in reducing mortality among psychiatric patients after discharge. It has broader significance too, opening the question of whether such assistance could reduce the well established high mortality rate among all people with psychiatric disorders.

Overall, the study appears to be solid, once one understands what the authors did. The presentation, however, is somewhat confusing. It took us quite some time to understand what they did. For most readers, who will not be familiar with the 100 million cohort nor the BFP, it would be much more difficult. In that regard we have several comments: 3 main ones and then several smaller ones. We have not included the many ways in which the study could be expanded in scope or in detail, because we recognize that the paper is already reporting an important result, and further results could be in future papers.

Main Comments

First

The authors need to clarify the study sample in the text and in Figure 1. The large number who were excluded due to registering for CadUnico after 2015 is not noted in the text under Participants where Figure 1 is first referred to. In Results, where the authors again refer to Figure 1, they only note that they excluded persons who did not meet inclusion criteria, without noting that the largest number were excluded due to registration after 2015 (this can be found in the Figure 1 but not easily). It is not clear why these persons were considered participants in the first place, but whatever they call them, what is important is to describe the study sample upfront and highlight the key points clearly, such as this one. 

Note in the manuscript that the profile of individuals hospitalized for psychiatric reasons in other studies differs markedly from the cohort studied here (see Rocha et al, 2021, Revista de saude publica). The authors should also note how that might be due to the selection criteria for this study.

Other points that need to be clarified about the study sample include: 

were these "first ever" psychiatric hospitalizations, or "first during 2008-2015"? 

what were the main reasons why some who applied for BFP received it or did not receive it? 

for those who received BFP, how long was the period between hospitalization and receipt? 

what proportion were in psychiatric versus general hosptials? were there differences in mortality between these two groups? between those with primary vs secondary psychiatric diagnoses?

When the answers to these questions are not available, that should simply be stated.

Second

In the introduction or discussion, it would be helpful to further situate the reader in terms of the unified healthcare system in Brazil, who it serves, and the general psychosocial attention policies (e.g., Law 10.216/2001, implementation of CAPS, reduction of hospital beds in psychiatric hospitals and increase in beds in general hospitals). 

Also, clarify how individuals who are registered in the CadUnico differ from the entire population even though they comprise over 60% of Brazil's population. One sentence referencing the original 100 million cohort paper should suffice to situate the reader. 

Line 364: this may not be clear to international readers. A potential alternative "community based mental health centers (called psychosocial care centers, or CAPS, in Brazil)"

Space could be made by implementing the third suggestion below.

Third 

The main results are in Tables 3 and 4, and Table 2 is not needed for the main text. Since Table 2 has only unadjusted results, it should be made a supplement. Table 2 provides information that might be needed to fully evaluate the study. Anything that needs to be highlighted- such as differences between BFP and non-BFP in person-years at risk- could be briefly alluded to in the text. However, presenting the unadjusted results in detail in the main text and table 2, before the main results, is quite confusing for readers. 

Since groups differ on person-years at risk, could the authors explain the choice of IRRs rather than doing a survival analysis such as proportional hazards? We don't insist on doing a survival analysis, but think some rationale for the approach chosen is needed.

Smaller points:

Line 84: we think mortality risk should be mortality rate.

Line 112: note lack of LMIC data on mortality for psychiatric disorders.

Line 113: "approximately 62·2% of the global burden of suicide deaths may be attributed to psychiatric disorders". This statement derives from a paper but might be problematic because "attributing" suicide deaths to a single factor across the globe is fraught with problems. It seems unnecessary here though we leave this to the authors' discretion.

Lines 114-116: include the role of psychiatric treatments in mortality (e.g., long term neuroleptic use)

Line 131: note that the post-discharge period is particularly vulnerable, and that psychiatric hospitalizations themselves may increase all-cause mortality, as shown in many articles.

Line 237 delete statement that SMDS >.1 indicated confounding. This would not be the correct approach, and indeed, they do not follow it in constructing the propensity scores. 

Line 260: how was length of hospitalization obtained? Authors may be aware that the typical way to obtain that information is not reliable because administrators often circumvent the need to maintain the same AIH number for hospitalizations longer than 15 days and often generate a new number, which may cause problems calculating length of stay. Other authors have found ways to address this, please clarify how you dealt with this issue or if LOS was calculated in a different way

Line 281: please provide rationale for excluding everyone who received Bolsa Familia before their hospitalization, as it seems that this would be an important group to examine- but might be beyond the scope of the study. 

Line 305: interesting finding, could there be group differences because women were more likely to have been excluded from this cohort for being on benefits before hospitalization (e.g., because of pregnancy)? 

Line 331: change "there were effects" to "results were consistent with an effect" 

Line 365: consider explaining that a national policy gradually reduced the numbers of hospital beds in psychiatric hospitals and implemented community based care nationally starting in 2021. 

Lines 367-369: the national policy encourages hospitalizations in general hospitals and discourages psychiatric beds in psychiatric hospitals. Authors may be referring to the fact that smaller cities may not have professionals with the right expertise (psychologists, psychiatrists, occupational therapists, psychiatric nurses), or the appropriate space in general hospitals to serve the complex needs of this population

Lines 455-456: national policy encourages general hospital admission vs psychiatric hospital, please revise. 

Ezra Susser and Ana Florence

[LINK]

COMMENTS FROM THE ACADEMIC EDITOR:

The Academic Editor was supportive of your work from the outset. She commented that she believed that you address a very important topic and also commented positively regarding your unique data set. She noted the concerns raised by the statistical reviewer, and encourages you to address these.

GENERAL EDITORIAL REQUESTS:

1. Please upload any figures associated with your paper as individual TIF or EPS files with 300dpi resolution at resubmission; please read our figure guidelines for more information on our requirements: http://journals.plos.org/plosmedicine/s/figures. While revising your submission, please upload your figure files to the PACE digital diagnostic tool, https://pacev2.apexcovantage.com/. PACE helps ensure that figures meet PLOS requirements. To use PACE, you must first register as a user. Then, login and navigate to the UPLOAD tab, where you will find detailed instructions on how to use the tool. If you encounter any issues or have any questions when using PACE, please email us at PLOSMedicine@plos.org.

To submit your revised manuscript please use the following link:

---

## [Decision Letter · Decision Letter 2]

23 Aug 2024

Dear Dr Bonfim,

Many thanks for submitting your revised manuscript "Do conditional cash transfers reduce mortality in people hospitalised with psychiatric disorders? A cohort study of the Brazilian Bolsa Família Programme" (PMEDICINE-D-24-01060R2) to PLOS Medicine. Firstly, let me say that we are grateful for your thorough engagement with the reviewer comments and appreciate the efforts made to act on these. The paper has now been reviewed again by a subject expert and a statistician; their comments are included below and can also be accessed here: [LINK]

As you will see, the reviewers were largely positive about the revised paper, but the statistician had a number of recommendations. After discussing the paper with the editorial team and an academic editor with relevant expertise, I'm pleased to invite you to revise the paper in response to the statistician’s comments. Please note that we plan to send the revised paper back to the statistical reviewer, and we cannot provide any guarantees at this stage regarding publication.

When you upload your revision, please include a point-by-point response that addresses all of the reviewer and editorial points, indicating the changes made in the manuscript and either an excerpt of the revised text or the location (e.g.: page and line number) where each change can be found. Please also be sure to check the general editorial comments at the end of this letter and include these in your point-by-point response. When you resubmit your paper, please include a clean version of the paper as the main article file and a version with changes tracked as a marked-up manuscript. It may also be helpful to check the guidelines for revised papers at http://journals.plos.org/plosmedicine/s/revising-your-manuscript for any that apply to your paper.

We ask that you submit your revision by Sep 13 2024 11:59PM. However, if this deadline is not feasible, please contact me by email, and we can discuss a suitable alternative.

Don't hesitate to contact me directly with any questions (ssunny@plos.org). 

Best regards, 

Syba 

Syba Sunny, MBBS, MRes, FRCPath 

Associate Editor

PLOS Medicine

ssunny@plos.org

Comments from the academic editor:

The academic editor continued to be supportive of your manuscript and hopes that you can address the statistician’s points.

Comments from the reviewers: 

Reviewer #1 (statistician):

I thank the authors for their responses to my comments on the previous version of this manuscript. However, I have noticed some inconsistencies throughout the manuscript and supplementary materials that require attention. Please note that line numbers in my comments refer to those on the tracked changes version of the manuscript.

1. Abstract: please change "competitive risk" to "competing risks" (here and on line 356 and Tables 2, 3, and in the Supplementary material)

2. Line 315: IPTW estimation is mentioned before it is defined. Please define this acronym when first used.

3. Is the primary aim to estimate the ATT (average effect in the treated), as stated on line 347? This should be stated early on in the Statistical Analyses section. If this is the case the authors were correct in their previous choice of weights: 1 for those individuals who did receive a cash transfer, and PS/(1-PS) for those who did not. However, I am confused since on line 389 it is stated that estimation of the ATT is done as sensitivity analysis (or is this just referring to one particular method for estimation of the ATT)? I also note that the text in the manuscript regarding the weights should match that in the Appendix. 

4. Are the SMDs in Table 1 for weighted or unweighted data? Please provide SMDs for the unweighted and weighted data in Table 1. Why do the SMDs in Table 1 not match those in the before IPTW column in S4 Table 4?

5. S4 Fig 3 does not match the data provided in S4 Table 3 - for example, the maximum propensity score in the non-BFP sample is stated as being 0.767 in the table, but is >0.8 in the figure. 

6. Please reduce the number of significant figures provided in the tables in S5 to 2.

7. An intraclass correlation is provided in S6. What is the outcome that this is calculated for? Were multiple members of households hospitalised with psychiatric disorders? How many households were included, and what sizes were these? Is this referred to anywhere in the paper? I would suggest deleting this section.

Reviewer #3: 

My comments have been addressed satisfactorily. I have no further comments.

---

* Please upload any figures associated with your paper as individual TIF or EPS files with 300dpi resolution at resubmission; please read our figure guidelines for more information on our requirements: http://journals.plos.org/plosmedicine/s/figures. While revising your submission, please upload your figure files to the PACE digital diagnostic tool, https://pacev2.apexcovantage.com/. PACE helps ensure that figures meet PLOS requirements. To use PACE, you must first register as a user. Then, login and navigate to the UPLOAD tab, where you will find detailed instructions on how to use the tool. If you encounter any issues or have any questions when using PACE, please email us at PLOSMedicine@plos.org.

* Thank you for providing a data availability statement. We have a few requests for amendments here.

(1) Please note that a study author cannot be the contact person for the data. We ask that details be provided so that data requests can be made to a non-author institutional point of contact, such as a data access or ethics committee, as this helps guarantee long term stability and availability of data. Providing interested researchers with a durable point of contact ensures data will be accessible even if an author changes email addresses, institutions, or becomes unavailable to answer requests. 

(2) We note that there are appears to be 3 datasets that have been accessed in this study. Could you provide access details for all three datasets (in the relevant metadata section) please? 

* We note that the datasets referred to in your study have been used by other publications. Nevertheless, it might be useful for readers if you could perhaps provide a short summary of the datasets in the Supporting Information files? This may enable the reader to understand the nature of the data a bit better without having to leave the article page.

* Thank you for including the STROBE checklist in your Supporting Information. Could you kindly revise this so that you only use section and paragraph numbers, rather than page numbers? This is because page numbers can change at a later point in the publication process. Please also add the following statement, or similar, to the Methods: "This study is reported as per STROBE guideline (S1 Checklist)."

SUPPLEMENTARY MATERIAL

REFERENCES

---

## [Decision Letter · Decision Letter 3]

26 Sep 2024

Dear Dr. Bonfim,

Thank you very much for re-submitting your manuscript "Do conditional cash transfers reduce mortality in people hospitalised with psychiatric disorders? A cohort study of the Brazilian Bolsa Família Programme" (PMEDICINE-D-24-01060R3) for review by PLOS Medicine.

I have discussed the paper with my colleagues and the academic editor and it was also seen again by the statistical reviewer. There are some outstanding requests from the statistical reviewer. However, I am pleased to say that provided the remaining requests are satisfied, and editorial and production issues are dealt with, we are planning to accept the paper for publication in the journal.

The remaining issues that need to be addressed are listed at the end of this email. 

In revising the manuscript for further consideration here, please ensure you address the specific points made by the reviewer and the editors. In your rebuttal letter you should indicate your response to the reviewers' and editors' comments and the changes you have made in the manuscript. Please submit a clean version of the paper as the main article file. A version with changes marked must also be uploaded as a marked up manuscript file.

We expect to receive your revised manuscript within 2 weeks (extended from the usual 1 week to take into account the statistical reviewer’s requests). Please email us (ssunny@plos.org) if you need more time.

We look forward to receiving the revised manuscript by Oct 10 2024 11:59PM. 

Sincerely,

Syba

Syba Sunny, MBBS, MRes, FRCPath

Associate Editor 

PLOS Medicine

ssunny@plos.org

Comments from Statistical Reviewer:

Reviewer #1: I thank the authors for their responses to my comments on the previous version. My only remaining comment is to do with the intracluster correlation and S6 text. I appreciate the additional context the authors provided regarding the inclusion of this material, and agree that it should remain in the Supplementary material. However, the issue is not the intracluster correlation itself (and I note that an ICC of around 0.02 as found here is meaningful and can have a large impact on inference for the outcome). The issue is to do with the estimation of the effect of the exposure when clustering is accounted for in this way - please include the estimated exposure effect and 95% confidence interval for this fitted model in the main paper. That is, the sentence "Our sample had no clustering effects" should be deleted, while a sentence like "When clustering by family was accounted for, the estimate of the effect of BFP on OUTCOME was ??? (S6 Text)" should be included in an appropriate place in the Results section. Please also state in the S6 text which model had mixed effects included: was this the Poisson model for mortality rates per 100,000 person years?

Comments from the editor:

Thank you for engaging with the review process so thoroughly. I have some very minor requests:

1) Abstract - Methods and Findings: Please expand the abbreviation ‘ICD-10’.

2) I note that you’ve used the abbreviation ‘CCTP’ for conditional cash transfer programmes, yet it’s referred to as ‘CTP’ in the abstract. Could you revise for consistency across the abstract and main text please?

3) Line 193 of clean revised copy: The first instance of the abbreviation ‘BFP’ is not expanded. (Rather I see that the 2nd instance on line 195 is.) Could you kindly expand on the first instance that ’BFP’ appears in the main text please?

---

## [Decision Letter · Decision Letter 4]

9 Oct 2024

Dear Dr Bonfim, 

On behalf of my colleagues and the Academic Editor, Charlotte Hanlon, I am pleased to inform you that we have agreed to publish your manuscript "Do conditional cash transfers reduce mortality in people hospitalised with psychiatric disorders? A cohort study of the Brazilian Bolsa Família Programme" (PMEDICINE-D-24-01060R4) in PLOS Medicine.

PRESS

Sincerely, 

Syba

Syba Sunny, MBBS, MRes, FRCPath 

Associate Editor 

PLOS Medicine